# Exopolysaccharides Producing Bacteria: A Review

**DOI:** 10.3390/microorganisms11061541

**Published:** 2023-06-09

**Authors:** Alexander I. Netrusov, Elena V. Liyaskina, Irina V. Kurgaeva, Alexandra U. Liyaskina, Guang Yang, Viktor V. Revin

**Affiliations:** 1Department of Microbiology, Faculty of Biology, M.V. Lomonosov Moscow State University, 119234 Moscow, Russia; 2Faculty of Biology and Biotechnology, High School of Economics, 119991 Moscow, Russia; 3Department of Biotechnology, Biochemistry and Bioengineering, National Research Ogarev Mordovia State University, 430005 Saransk, Russia; irina.kurgaewa@yandex.ru (I.V.K.); revinvv2010@yandex.ru (V.V.R.); 4Institute of the World Ocean, Far Eastern Federal University, 690922 Vladivostok, Russia; ochotona.hyperborea221b@yandex.ru; 5Department of Biomedical Engineering, College of Life Science and Technology, Huazhong University of Science and Technology, Wuhan 430074, China; yang_sunny@yahoo.com

**Keywords:** bacterial exopolysaccharides, bacterial strains, xanthan, levan, bacterial cellulose, environmental remediation

## Abstract

Bacterial exopolysaccharides (EPS) are essential natural biopolymers used in different areas including biomedicine, food, cosmetic, petroleum, and pharmaceuticals and also in environmental remediation. The interest in them is primarily due to their unique structure and properties such as biocompatibility, biodegradability, higher purity, hydrophilic nature, anti-inflammatory, antioxidant, anti-cancer, antibacterial, and immune-modulating and prebiotic activities. The present review summarizes the current research progress on bacterial EPSs including their properties, biological functions, and promising applications in the various fields of science, industry, medicine, and technology, as well as characteristics and the isolation sources of EPSs-producing bacterial strains. This review provides an overview of the latest advances in the study of such important industrial exopolysaccharides as xanthan, bacterial cellulose, and levan. Finally, current study limitations and future directions are discussed.

## 1. Introduction

Bacterial exopolysaccharides (EPSs) have recently received significant attention due to their unique structure and properties and the prospects for use in various fields of science, industry, medicine, and technology [1,2,3,4,5,6]. The term exopolysaccharide was first introduced by Sutherland in 1972 [7] for high molecular weight carbohydrate biopolymers produced by microorganisms. Bacterial EPSs have a diverse structure and are secreted by a wide range of bacteria [8]. Depending on their localization, bacterial EPS are divided into capsular polysaccharides that are closely associated with the cell surface, and free slime polysaccharides loosely attached or even totally secreted into the extracellular environment [3,9]. EPSs synthesized by bacteria have an advantage over those isolated from plants (cellulose, starch, and pectin), animals (glycogen and chitin), and algae (agar, fucoidan, carrageenans, and alginates). Bacterial EPSs can be obtained regardless of the season and weather conditions on an industrial scale. They are extracellular slime that is easily released from cells into the environment. Therefore, the methods for their extraction from the cell-free supernatant are quite simple and cost-effective. Bacteria multiply rapidly and are characterized by metabolic flexibility and a variety of physiological and biochemical properties. By means of the methods for optimizing cultivation conditions, genetic and metabolic engineering, it is possible to modulate the yield as well as the structural and functional properties of bacterial EPSs [9,10,11,12]. Bacterial EPS are characterized by the presence of a large number of functional groups (hydroxyl, carboxyl, carbonyl, acetate, etc.), that enable them to modify their molecules in order to impart new valuable properties to them [13]. Therefore, recent interesting reviews by Aditya et al. and Aziz et al., have presented the modification of bacterial cellulose (BC) using chemical and physical methods to obtain nanocomposites and fabricate materials with improved functionality for biomedical applications [13,14]. Generally, EPS are classified into two types: homopolysaccharides, which are either unbranched or branched and composed of single-type monosaccharides such as glucose and fructose linked through glycosidic bonds, and heteropolysaccharides, which contain two or more units of different monosaccharides (glucose, fructose, galactose, mannose, rhamnose, fucose, N-acetylglucosamine, and uronic acids) [4,8]. Homopolysaccharides are divided into α-D-glucans (dextran, alternan, and reuteran), β-D-glucans (bacterial cellulose), fructans (levan and inulin), and polygalactans. Heteropolysaccharides are polymers such as xanthan, alginate, hyaluronic acid, kefiran, and gellan [4,15]. EPSs biosynthesis in bacteria occurs intra- and extracellularly. There are four general mechanisms for EPS biosynthesis in bacterial cells; they are the Wzx/Wzy-dependent pathway, the ABC transporter-dependent pathway, the synthase-dependent pathway, and extracellular biosynthesis by sucrase protein [2,4]. Homopolysaccharides are usually synthesized using the synthase and extracellular synthesis pathways, while heteropolysaccharides are synthesized by the Wzx/Wzy-dependent pathway and the ABC transporter-dependent pathway.

EPS-producing bacteria belong to different phylogenetic groups and include Gram-negative bacteria of such classes as the *Alphaproteobacteria* class including *Acetobacter*, *Gluconobacter*, *Gluconacetobacter*, *Komagataeibacter*, *Kozakia*, *Neoasaia*, *Agrobacterium*, *Rhizobium*, and *Zymomonas* genera; the *Betaproteobacteria* class including *Alcaligenes* and *Achromobacter* genera; and the *Gammaproteobacteria* class including *Azotobacter*, *Pseudomonas*, *Enterobacter*, *Alteromonas*, *Pseudoalteromonas*, *Xanthomonas*, *Halomonas*, *Erwinia*, *Vibrio*, and *Klebsiella* genera. They also include Gram-positive bacteria of such classes as *Bacilli* including *Bacillus*, *Paenibacillus*, *Lactobacillus*, *Leuconostoc*, and *Streptococcus* genera; class *Clostridia* including *Sarcina* sp.; and class *Actinomycetia* including *Bifidobacterium*, *Rhodococcus* genera, and others [2,16,17]. Some of the most commonly used EPSs are xanthan from the genus *Xanthomonas*, dextran from the *Leuconostoc*, *Streptococcus*, and *Lactobacillus* genera, alginate from the *Azotobacter* and *Pseudomonas* genera, curdlan from *Alcaligenes faecalis*, *Rhizobium radiobacter*, and *Agrobacterium* sp., gellan from the *Sphingomonas* and *Pseudomonas* genera, hyaluronan from *Streptococcus* sp., levan from *Bacillus* sp., *Paenibacillus* sp., *Halomonas* sp., and *Zymomonas* sp., bacterial cellulose from *Komagataeibacter* sp., and others. There are also known polysaccharides such as fucogel produced by *Klebsiella pneumoniae*, clavan produced by *Clavibacter michiganensis*, fucoPol produced by *Enterobacter* sp., and kefiran produced by *Lactobacillus kefiranofaciens* [6]. The importance attached to the commercial applications of EPSs contributes to the rapid search for new producers’ isolation, characteristics, and the production of new EPSs in order to obtain novel functional materials with a wide range of applications. Therefore, novel EPSs from extremophilic and marine bacteria have attracted researchers’ attention for their potential to be used in various fields including medicine, food, environmental protection, etc. [18,19]. In addition, recently, researchers have paid significant attention to EPSs produced by probiotic bacteria (*Lactobacillus*, *Lactococcus*, *Bifidobacterium*, *Streptococcus,* and *Enterococcus*) for various applications, particularly in medicine [15,20,21,22,23,24].

The biological role of bacterial EPSs is diverse. They protect cells against extreme temperature [25,26], salinity [27], aridity [28], UV-rays, unfavorable pH values, osmotic stress, phagocytosis, and chemical agents (antibiotics, heavy metals, and oxidants) [2]. Most marine bacteria secrete polysaccharides, which are important for their survival in the marine environment [25]. EPSs have a cryoprotective effect, in particular, for arctic bacteria [26]. The bioethanol producer *Zymomonas mobilis* is able to grow in media with an alcohol concentration of up to 16% that can be explained by the protective effect of the EPSs formed by it; the first of them consists of mannose, and the second contains galactose [29]. EPSs play an important role in bacterial adhesion, aggregation, and biofilm formation and are the main fraction of the biofilm matrix both in Gram-positive and Gram-negative bacteria [30,31,32]. EPSs of the biofilm matrix promote horizontal gene transfer, and intercellular interactions prevent the dehydration of bacteria and provide protection against external agents including antibiotics. Therefore, biofilm formation is associated with higher antibiotic resistance and is an important surgical problem. Understanding the composition of biofilms and each component function is crucial to the development of new therapeutics against infections caused by pathogens (*Staphylococcus aureus*, *Klebsiella pneumoniae*, *Pseudomonas aeruginosa*, *Enterococcus faecium*, and *Enterobacter* sp.). Therefore, the review by Balducci et al. (2023) described the structure and the role of different EPSs in bacterial biofilms of pathogens and provided an overview of potentially novel antimicrobial therapies capable of inhibiting biofilm formation by targeting EPS [32]. EPS plays an important role in controling virulence not only of human pathogens but also those of plants. Xanthan produced by the phytopathogenic bacteria *Xanthomonas* spp. is one of the virulence factors involved in the specific interaction of bacteria with the plant [33,34]. EPSs of pathogenic *Agrobacterium* species are also involved in the adhesion of bacterial cells during infection [35,36].

The growing number of publications devoted to bacterial EPSs in the last decade is primarily related to the prospects for their practical application (Figure 1). Bacterial EPSs have many unique beneficial properties such as biocompatibility, biodegradability, non-toxicity, the ability for gelation, high adhesive ability, viscoelasticity, pseudo-plasticity, and thixotropic nature. EPS also showed the potential to withstand various environmental stresses such as elevated temperature, extreme pH, freeze-thaw, and high salt concentrations. Therefore, they have extensive commercial applications in food, pharmaceutical, cosmetic, chemical, textile, oil, and gas industries as thickeners, emulsifiers and suspension stabilizers, flocculants, and additives improving the quality of various products [37]. In addition, some bacterial EPSs also possess antitumor [38,39,40,41], antioxidant [41,42], anti-inflammatory, antibacterial, antiviral, cholesterol-lowering, prebiotic [43], wound healing, and immunomodulatory activities **[44]**. The biocompatibility and functional properties of EPSs are important factors that promote their use in various biomedical applications [45], such as tissue engineering [46,47], wound dressing [48,49], and drug delivery systems [50,51]. EPSs have shown good biocompatibility, biodegradability, and mechanical strength, which are beneficial to form biological scaffolds [43]. EPSs are potential carriers for valuable medicine, including growth factors and antitumor drugs **[52,53,54]**. Antibiotics are widely used as a model for drug delivery release with bacterial EPSs [55]. New bacterial polysaccharides have been obtained for the treatment of Alzheimer’s disease [56] and diabetes [51]. EPSs can also be used for wastewater treatment from heavy metals and organic pollutants including dyes, pharmaceutical compounds, and petroleum products [57,58,59,60,61,62]. EPSs are attracting special attention as biopolymers to produce new biocomposite materials for a wide range of applications. The composites based on EPSs can be given antibacterial, wound healing, conductive, magnetic, optical, and other properties [14,63,64,65].

Despite the unique properties of bacterial EPS, and great prospects for their practical application, the number of them that are industrially produced is extremely limited. This is primarily due to the low productivity of bacterial strains, which have not yet reached the industrial application level, and, consequently, the high cost of the resulting products. In general, bacterial EPSs are formed in an amount from 0.29 to 65.27–100 g/L depending on their type, the type of microorganism, and the cultivation conditions. The maximum EPSs yield is observed in the producer of levan (up to 100 g/L) [66], dextran (up to 66 g/L) [67], kurdlan (up to 48 g/L) [67], and xanthan (up to 40 g/L) [67]. The bacteria forming bacterial cellulose and hyaluronic acid have low productivity (usually not more than 10 g/L) [68]. Therefore, further research is needed to set up a highly efficient production of EPSs. The main ways to reduce costs include cheap culture media usage, the isolation of novel highly productive strains, and the creation of more productive strains using genetic engineering. Genetic and metabolic engineering strategies are currently employed to increase EPSs production. In addition, EPSs production can be increased through the development and improvement of technological processes. The yield, structure, and physical-chemical properties of EPSs depend on many factors such as the cultivation condition, source of carbon, C/N ratio, pH of media, and cultivation time. The cultivation conditions of the producer (temperature, pH, amount of oxygen, and bioreactor type) have a significant effect on EPS biosynthesis. It is generally accepted that the production of most bacterial EPS requires a high content of a carbon source in the medium and limited nitrogen nutrition. To obtain bacterial EPS, the main carbon sources that are used are glucose and sucrose. Since the cost of culture media is about 30% of the cost of the entire fermentation process, a large number of media are currently offered based on industrial and agricultural waste [68]. Biocatalytic technologies are also a promising direction for obtaining bacterial EPS. An interesting review by Efremenko et al. (2022) presents recent results and achievements in biocatalysis [67]. The authors note that a common feature of all these catalysts in such processes is an increased concentration of cells and their transition to a quorum sensing (QS) state. QS provides the activation of EPSs synthesis [69,70] as protective, stabilizing, and reserve substances for highly concentrated microbial populations, which is a natural mechanism to increase the amount of these biopolymers and can be used as a nature-like technology in their industrial production. Therefore, the aim of this review was to summarize the current research progress on bacterial EPSs with special attention paid to the bacterial cellulose-, xanthan-, and levan-producing bacteria.

## 2. Bacterial Cellulose-Producing Bacteria

Cellulose is the most abundant polymer on earth and is primarily produced by plants; however, some microorganisms are also the producers of this polysaccharide. Bacterial cellulose (BC) has attracted much attention over the last years as a unique biomaterial with ultrafine fibrous network architecture and exceptional physicochemical and mechanical properties including high purity, high surface area, high polymerization degree (up to 8000), and high crystallinity (up to 90%) as well as superior mechanical properties (Young’s modulus about 15–35 GPa and tensile strength of 200–300 MPa), high water-holding capacity, lightweight, transparency, flexibility, good biocompatibility, good biodegradability, renewability, and non-toxic, non-immunogenic features [12,71,72,73,74,75]. Furthermore, most properties of BC are superior to plant-derived cellulose. Among others, BC fibers are about 100 times thinner than plant-derived cellulose (20–100 nm) and can hold water that exceeds a hundredfold their dry weight [12]. Bacterial celluloses’ unique properties are highly dependent on bacterial species.

BC is produced by Gram-negative bacteria of the genera *Komagataeibacter* (*Gluconacetobacter*) [76,77], *Acetobacter* [78], *Gluconobacter* [79], *Agrobacterium* [80], *Achromobacter* [81], *Enterobacter* [82,83], *Rhizobium* [84], *Pseudomonas* [85], *Salmonella* [86], and others, as well as Gram-positive bacteria of the genera *Bacillus* [87], *Sarcina*, and *Rhodococcus* [88]. The most common and highly productive BC producers are acetic bacteria species of the *Komagataeibacter* genus such as *K. xylinus* and *K. hansenii*. The *Komagataeibacter* genus belongs to the *Acetobacteraceae* family, class *Alphaproteobacteria*, phylum *Proteobacteria*. It was named after the famous Japanese microbiologist Dr. Kazuo Komagata, a Professor at the University of Tokyo, who made a great contribution to the taxonomy of bacteria, especially acetic bacteria. *Komagataeibacter* genus was separated from the plant-associated *Gluconacetobacter* on the basis of a 16S rRNA gene sequence and several morphological and physiological properties such as the inability to produce 2,5-diketo-d-gluconate as well as water-soluble brown pigment from glucose [89,90,91]. As of April 2023, the List of Prokaryotic names with Standing in Nomenclature (LPSN) includes the *Komagataeibacter* genus containing 19 species: *K. cocois*, *K. diospyri*, *K. europaeus*, *K.hansenii*, *K. intermedius*, *K. kakiaceti*, *K. kombuchae*, *K. maltaceti*, *K. medellinensis*, *K. melaceti*, *K. melomenusus*, *K. nataicola*, *K. oboediens*, *K. pomaceti*, *K. rhaeticus*, *K. saccharivorans*, *K. sucrofermentans*, *K. swingsii*, and *K. xylinus*. Table 1 presents the type strains of the species of *Komagataeibacter* genus, the sources of their isolation and general properties of the genome sequences [90,91,92,93,94,95,96,97,98,99]. In addition, complete genome sequences were obtained of the following strains of the genus *Komagataeibacter*: *K. xylinus* NBRC 3288 [100], *K. nataicola* RZS0111 [101], *K. hansenii* ATCC 53582 [102], *K. xylinus* E25 [103], *K. xylinus* CGMCC 2955 [104], *K. xylinus* E26, and *K. xylinus* BCRC 12334 [105], *K. xylinus* CGMCC 17276 [106], *K. rhaeticus* ENS9b [107], *K. rhaeticus* ENS 9a1a [108], *K. nataicola* RZS01 [109], *K. saccharivorans* JH1 [109], *K. europaeus* GH1 [110], and *K. intermedius* ENS15 [111]. All *Komagataeibacter* genomes are characterized by a high % GC content, which is the highest for *K. rhaeticus* LMG 22126^T^ (63.5%) and the lowest for *K. hansenii* JCM 7643^T^ (59.3%) [112].

Cellulose-producing bacterial cells of the genus *Komagataeibacter* are Gram-negative, rod-shaped, and single or paired; some of them are in the form of short chains, and their size is about 0.4–1.2 µm in width and 1.0–3.0 µm in length [77,113]. The colonies of cellulose-synthesizing strains are jelly-like, rounded, and uplifted in the center (Figure 2B) [77,114,115]. The dissolution of calcium carbonate on the Acetobacter agar plate indicates the presence of acid-forming bacteria. Numerous recently isolated BC producers belong to the genus *Komagataeibacter.* Many strains were isolated from the kombucha community: *K. kombuchae* LMG 23726^T^ [94], *K. hansenii* GH-1/2008 (VKPM B-10547) [116], *K. xylinus* B-12068 [117], *K. rhaeticus* P 1463 [118], *K. rhaeticus* K3 **[119]**, *K. intermedius* AF2 [120], *K. sucrofermentans* B-11267 (Figure 2) [77], and others. The kombucha community (*Medusomyces gisevii* J. Lindau) is a symbiotic culture of bacteria and yeast that is commonly abbreviated as SCOBY [121]. The microbial community in kombucha varies between geographic origin and cultural conditions [122,123]. According to the literature data, kombucha microflora may contain over 22 species including acetic acid bacteria of the family *Acetobacteraceae* and lactic acid bacteria of the genus *Lactobacillus*, as well as yeasts *Zygosaccharomyces* spp., *Saccharomyces* spp., *Dekkera* spp., and *Pichia* spp. [124]. Recently, a number of studies have been performed using metagenomics, comparative genomics, synthetic community experiments, and metabolomics to quantify community microbial diversity [122,125,126]. Therefore, Landis et al. (2022) determined the taxonomic, ecological, and functional diversity of 23 distinct kombuchas in the United States and demonstrated the bacterium *Komagataeibacter rhaeticus* and the yeast *Brettanomyces bruxellensis* to be the most common microbes in these communities [122]. In addition, Kahraman-Ilikkan showed that the dominant bacterium in kombucha samples from Turkey was *Komagataeibacter obediens*, and the dominant fungus was *Pichia kudriavzevii*, and also found propionic and butyric acid-producing bacteria such as *Anaerotignum propionicum* and *Butyrivibrio fibrisolvens* [123].

*Komagataeibacter* strains are highly acetic acid resistant (15–20%) and the dominant species in vinegar production processes [127]. Therefore, many strains were isolated from vinegar including *Komagatabacter* (*Gluconacetobacter*) sp. RKY5 [128], *K. medellinensis* LMG 1693^T^ [129], *K. medellinensis* [130], *K. europaeus* LMG 20956 [130], *K. melaceti* AV382^T^ [99,130], *K. hansenii* DSM 5602^T^ [130], and others. Therefore, Marič et al. (2020) isolated the two novel strains AV382 and AV436 from a submerged industrial bioreactor apple cider vinegar production in Kopivnik (Slovenia) [99]. These strains represent novel species of the genus *Komagataeibacter*, with the names *K. melaceti* sp. nov. and *K. melomenusus* being proposed for them, respectively. The type of strain of *K. melaceti* is AV382^T^ (=ZIM B1054^T^ = LMG 31303^T^ = CCM 8958^T^) and that of *K. melomenusus* is AV436^T^ (=ZIM B1056^T^ = LMG 31304^T^ = CCM 8959^T^). Recently, Ni et al. (2022) conducted a study with three new strains isolated from rice vinegar, among which *Acetobacter pasteurianus* MGC-N8819 showed a relatively high BC yield of 6.6 g/L on the Hestrin–Schramm (HS) medium [78]. Greser and Avcioglu (2022) isolated the strain *K. maltaceti* from grape vinegar and the strain *K.nataicola* from apple vinegar, which formed BC in amounts of 6.45 g/L and 5.35 g/L, respectively [131]. *K. intermedius* [132], *G. swingsii* [133], *K. rhaeticus* DSM 16663^T^ [130], *G. rhaeticus* [133], *Komagatabacter* (*Gluconacetobacter*) sp. gel_SEA623-2 [134], *Gluconacetobacter* sp. F6 [135], *K. maltaceti* SKU 1109 [130], *K. diospyri* MSKU 9^T^ [93], and *K. saccharivorans* JH1 [109] were isolated from fruit and fruit juices; *K. cocois* WE7^T^ was isolated from coconut milk [92]; *K. nataicola* LMG 1536^T^ was isolated from “Nata-de Coco” [130]; and *K. hansenii* B-12950 was isolated from the Tibicos symbiotic community [77]. A new species of thermotolerant bacterium *Komagataeibacter diospyri* sp. nov. designated MSKU 9^T^ was isolated from persimmon [93].

BC plays an important role in bacterial physiology and ecology [136,137,138,139,140,141,142]. For example, BC film on the medium surface ensures the maintenance of an aerobic environment [141]. The water-holding capacity of vegetable cellulose reaches 60%, while the water-holding capacity of BC is 100% of its dry weight. Thus, it protects the cells from drying out. BC is supposed to form a kind of “framework” protecting cells against external agents including antibiotics, heavy metal ions, and UV exposure [137]. BC producers can exhibit symbiotic or pathogenic relationships with plants, animals, or fungi. And in this case, BC serves as a kind of molecular glue to ensure interactions in nature [136]. For example, BC is involved in the adhesion of bacterial cells of the genus *Rhizobium* during symbiosis with leguminous plants, *Agrobacterium* and *Salmonella* in infection, contributing to the colonization of plants providing protection against competitors [35,84,86]. Cellulose and its derivatives are important components of biofilms and play a significant role in regulating the virulence of plant and human pathogens [32,137,138]. Biofilm formation is an effective strategy by which bacterial cells establish relationships in a unique environment [32]. The role of BC-containing biofilms is to establish close intercellular and host-bacteria interactions. A biofilm gives bacteria the ability to undergo horizontal gene transfer, which provides antibiotic resistance, inhibits bacterial dehydration, and protects them from extreme conditions such as mechanical stress and antibiotic treatment. Cell-to-cell communications in a biofilm proceed, among other things, with the participation of a regulatory mechanism called “quorum sensing” (QS). Information is exchanged using specialized chemical signaling molecules, through which the microbial community acts as a single organism [142]. Cyclic diguanylate (c-di-GMP) is an intracellular signaling molecule that regulates the transition from a planktonic state in the environment to a surface-associated biofilm for many bacteria including *Agrobacterium* sp., *Rhizobium* sp., *Salmonella* sp., Vibrio sp., *Pseudomonas* sp., and others [80,84,140,143]. The review by Augimeri et al. (2015) highlighted the diversity of BC biosynthesis and regulation and its role in environmental interactions by discussing diverse biofilm-producing *Proteobacteria* [136]. The review by Balducci et al. (2023) described the exopolysaccharides that are most commonly found in the biofilm matrix of pathogens causing infections in humans and their functions in forming and maintaining the EPS matrix. Biofilms are an important surgical issue since they can form on a chronic wound or implantable medical devices [32]. *E. coli* and *Salmonella* sp. are among the most studied pathogenic bacteria producing cellulose. Recently, *E. coli* and *Salmonella* species have been discovered to produce chemically modified cellulose with phosphoethanolamino groups. This modification seems to have multiple functions in the extracellular matrix: it is required to form long cellulose fibrils and a tight nanocomposite with curli fibers, it may confer resistance against attacks by cellulase-producing microorganisms, and it can prevent curli from hyper stimulating immune responses [144]. Finally, there are bacteria known to secrete cellulose; however, specific secretion mechanisms have not been identified so far. For example, pathogenic *Mycobacterium tuberculosis* has been shown to secrete biofilm-promoting cellulose both in vitro and in granulomatous lesions in the lungs of infected hosts in vivo [145]. These data suggest that the cellulose-rich extracellular matrix contributes to mycobacterial drug tolerance, simultaneously protecting the pathogen against triggering immune responses in the host. 

Despite the biological and practical significance of BC, the molecular mechanism of its biosynthesis is only just starting to emerge due to the improvement of next-generation sequencing technologies, the publication of the genome sequences of numerous BC producers, and the increased availability of genetic tools [112,137]. Recently, many reviews of the latest advances in structural and molecular biology in BC biosynthesis have been published. The review by McNamara et al. (2015) presented a detailed molecular description of cellulose biosynthesis [146]. The review by Li et al. (2022) showed the research progress of biosynthetic strains and pathways of bacterial cellulose [88]. Abidi et al. (2022) reviewed current mechanistic knowledge on BC secretion with a focus on the structure, assembly, and cooperativity of BCs secretion system components [147]. An interesting review by Manan et al. (2022) provided a comprehensive overview of the molecular regulation of the intracellular biosynthesis of cellulose nanofibrils, their extracellular transport, and their organization into highly ordered supramolecular structures [12]. The authors discussed in detail the role of different operons involved in BC biosynthesis and their regulation to achieve high yield and productivity through the genetic engineering of BC-producing strains. The review by Ryngajłło et al. (2020) discussed the current progress in the systemic understanding of *Komagataeibacter* physiology at the molecular level [112]. The authors presented examples of the approaches, as well as genetic engineering strategies for strain improvement in terms of BC synthesis intensification.

In general, bacterial cellulose biosynthesis comprises three steps including uridine diphosphate glucose synthesis through a series of enzymatic reactions, cellulose molecular chain synthesis under the function of cellulose synthase, and cellulose crystallization and polymerization. The cellulose synthase (CS) complex is encoded by cellulose synthase operons known as *bcs* operons. The *bcs* operons regulate intracellular biosynthesis, extracellular transport across the cellular membranes, and in vitro assembly of cellulose fibrils into highly ordered structures [12]. In Gram-negative bacteria, CS is comprised of a four-subunit transmembrane complex, where the BcsA, BcsB, and BcsC subunits are responsible for the synthesis and extracellular transport of glucan chains, while the fourth one, BcsZ (formerly known as BcsD), resides in the periplasm and performs the *endo*-β-1,4-glucanase activity. One of the well-characterized mechanisms regulating cellulose biosynthesis is the allosteric activation of BcsA with a cyclic di-GMP (c-di-GMP) molecule, a universal bacterial second messenger discovered in *K. xylinus* [112]. The independent research revealed an important regulatory role of c-di-GMP for motility, virulence, and biofilm formation [137]. An interesting article by Liu et al. (2018) presented a comprehensive approach to studying the molecular mechanisms of BC biosynthesis and metabolism regulation based on a complete genome analysis of *Gluconacetobacter xylinus* CGMCC 2955 and other BC producers whose genomes were sequenced completely [104]. A complete genome sequence is needed as background information for genetic engineering to achieve precise control of BC biosynthesis based on metabolic regulation. The authors made a comparison of the arrangement and composition of BC synthase operons (*bcs*) with those of other BC-producing strains. In addition, they demonstrated the presence of QS in *G. xylinus* CGMCC 2955 and proposed a possible regulatory mechanism action of QS on BC production [104].

Despite numerous studies conducted in recent years, the large-scale production of BC remains quite expensive. This is mainly due to the low productivity of bacterial strains, which, as a rule, do not exceed 5 g/L BC. According to the review by Li et al. (2022), the maximum BC yield did not exceed 20 g/L, which has not yet reached the level of industrial application [88]. Recently, several reviews on the genetic modification of bacterial strains for enhancing BC production were reported [148,149]. The review by Singhania et al. (2021) presented the mechanisms and targets for genetic modifications in order to achieve the desired changes in the BC production titer as well as its characteristics [148]. The authors note that the lack of studies on a genetic modification for BC production is due to the limited information on the complete genome and genetic toolkits; however, over the past few years, the number of studies in this area has increased, since the whole genome sequencing of several strains of *Komagataeibacter* has been performed. Genetic engineering enables the modification of the genetic material of *Komagataeibacter* to increase the product yield, reduce the risk of deleterious mutations, and improve or change cellulose properties such as crystallinity, mechanical strength, and porosity, which is suitable for specific applications. Along with the above-mentioned advantages, there are certain problems including methodological problems of transformation and the issues related to the regulatory process complexity, when each gene can express a protein performing more than one function [148]. However, there have been several attempts at genetic engineering for BC-producing bacteria. For example, Kuo et al., generated a *G. xylinus* mutant by knocking out the membrane-bound glucose dehydrogenase gene via homologous recombination of a defect in the gene, which led to the formation of cellulose from glucose without generating gluconic acid and a 40% increase in BC production [150]. Japanese scientists obtained the recombinant *E. coli* bacteria capable of forming BC resulting from the transfer of *G. xylinus* genes [151]. A new stable efficient plasmid-based expression system of recombinant BC in the *E. coli* DH5_ platform has currently been developed [152].

Furthermore, BC production can be increased through the development and improvement of technological processes, such as the optimization of the nutrient medium, culture conditions, cultivation methods, and the development of cell-free culture systems. About 30% of the total cost of the process is known to be the cost of the nutrient medium [153]. The most frequently used medium is the Hestrin–Schramm (HS) medium, which includes some expensive components, such as glucose, yeast extract, peptone, citric acid, and disodium phosphate, resulting in costly production. In order to reduce the cost, researchers attempted to produce cellulose from various alternative substrates, particularly, agro-wastes, pulp mills and lignocellulosic wastes, biodiesel industry wastes, acetone-butanol-ethanol fermentation wastewater, and others [68,154,155]. BC production also depends on a cultivation technique. Under static cultivation, bacteria form cellulose in the form of a film on the medium surface (Figure 2D). Under agitated conditions, most strains form cellulose in the form of agglomerates of various shapes depending on the medium composition and mixing modes (Figure 2G,H) [77,156,157,158]. There are many reports analyzing static and agitated conditions for cellulose production by *Komagataeibacter* as well as other bacteria, where a static culture generally gave higher yields compared to agitated cultures [159], but there are reports of higher BC yields with the agitated culture [160]. However, such a comparison is not always justified without taking into account the time and growing conditions. Furthermore, cell-free culture, the synthesis without living cells, shows great development prospects [161,162,163]. Cell-free culture could reduce the cost of BC production, decrease the metabolic inhibitors, and significantly improve the efficiency of enzymatic reactions [162]. In addition, an effective co-cultivation of *Komagataeibacter* with the producers of other polysaccharides was reported, which led to an increased BC yield, and can be used to obtain nanocomposites with improved functional characteristics [164].

Highly efficient BC production will enable the expansion of the scope of the polysaccharide with unique properties. BC is an attractive biopolymer for a number of applications including food, biomedical, cosmetics, and engineering fields. The review by Cubas et al. (2023) reported BC to be a new era in Green Chemistry products [165]. In the past five years, there have been many interesting, detailed reviews describing BC production, properties, and applications [43,65,68,71,72,73,74,76,166,167,168,169,170,171,172,173,174,175,176,177,178,179]. BC has great potential to be used in medicine [2,3,43,71,72,73,74,166,167,168,169,170,171,172,179] as a biomaterial for wound dressing [180,181,182,183,184,185,186], tissue engineering [187,188,189,190,191,192], and drug delivery systems [51,193,194,195,196,197]. In addition, BC can be used in the food industry as a nutritional component, food packaging, an emulsion stabilizer, and novel food [165,198,199,200]. BC is known as a fiber-rich natural food that offers many health benefits and reduces the risk of chronic diseases, such as diabetes, obesity, and cardiovascular diseases [199]. In 1992, BC was recognized safe by the Food and Drug Administration (FDA), and in 2019, the species *K. sucrofermentans* was included in the list of Qualified Presumption of Safety (QPS) recommended biological agents and intentionally added to food (novel food) [199]. Furthermore, BC can be used in the cosmeceutical industry [65,201,202]; the environmental industry as a matrix to immobilize catalysts, enzymes, and other sensory materials, both to detect environmental pollutants and also decompose various wastes from heavy metals, fluorine, organic pollutants including dyes, pharmaceutical compounds, and petroleum products [68,203,204,205,206]; in nanoelectronics (sensors, optoelectronic devices, flexible display screens, energy storage devices, and acoustic membranes) [167,207,208,209,210,211,212,213]; for microbial enhance of oil recovery [83]; in the textile industry [214]; and in biocatalytic technologies as a carrier for microorganisms or enzymes for various approaches [67,68,71].

## 3. Levan-Producing Bacteria

Levan is one of the most promising microbial EPSs for a wide range of biotechnological applications due to its beneficial functional properties such as anticancer [215,216,217], antioxidant [218], antibacterial [215,218], anti-inflammatory, immunomodulatory [219], and prebiotic activities [220], as well as unique physicochemical properties such as high adhesive strength, low intrinsic viscosity, high water solubility, film-forming ability, heat stability, and high biocompatibility [221,222,223]. Levan is a neutral homopolysaccharide composed of fructose units connected by β-2,6-glycoside bonds in the backbone and β-2,1 in its branches. 

Many bacteria are capable of synthesizing levan, including Gram-negative bacteria of the class *Alphaproteobacteria*, *Acetobacter*, *Gluconobacter* [224,225,226], *Komagataeibacter* (*Gluconacetobacter*) [227], and *Zymomonas* [228,229,230] genera, and those of the class *Gammaproteobacteria*, *Pseudomonas* [231], *Halomonas* [232,233], and *Erwinia* [234] genera, as well as Gram-positive bacteria of the class *Bacilli*: *Bacillus* [235,236,237,238,239,240,241,242,243,244,245], *Paenibacillus* [246,247,248,249,250,251,252,253,254,255,256,257,258,259,260], *Lactobacillus* [261], *Leuconostoc* [262] genera, etc. Currently, more than 100 bacteria species have been shown to produce levan [222]. The most studied levan producers among Gram-positive bacteria are the species of the class *Bacilli* including *Bacillus subtilis* [243,244,245], *Bacillus licheniformis* [241], *Paenibacillus polymyxa* [246,248,250,256,257,258,259,260], *Lactobacillus reuteri* [261], *Leuconostoc citreum* [262], and others. These bacteria were isolated from different sources. Therefore, the probiotic *Bacillus tequilensis*-GM was isolated from Tunisian fermented goat milk [235]. *B. siamensis* was isolated from fermented soybeans and was found to produce levan at high sucrose concentrations [236,237]. A new EPS-producing Gram-positive bacterium *B. paralicheniformis* was isolated from the rhizosphere of *Bouteloua dactyloides* (buffalo grass), which produced a large amount (~42 g/L) of levan having a high weight average molecular weight of 5.517 × 10^7^ Da [238]. Many species of marine bacteria of the genus *Bacillus* can synthesize levan. For example, a strain of *Bacillus* sp. SGD-03 produces a levan with a molecular weight of 1.0 × 10^4^ Da with a maximum yield of 123.9 g/L [239]. *B. paralicheniformis* ND2 was isolated from the seawater of the Mediterranean Sea, Egypt, followed by screening for levan-type fructan production, which yielded 14.57 g/L [240]. Levan produced by the *B. aryabhattai* GYC2-3 strain had a high average molecular weight (5.317 × 10^7^ Da) [242], while levan from *B. licheniformis* 8-37-0-1 had a low molecular weight (2.826 × 10^4^ Da) [241]. The sucrose concentration was found to be the critical component in modulating the molecular weight of synthesized levan. Using a low sucrose concentration (20 g/L) in a culture of *B. subtilis* (natto) Takahashi resulted in predominantly high molecular weight levan (>2 × 10^6^ Da). In contrast, low molecular weight levan (6–9 × 10^3^ Da) was the prevalent EPS when a high sucrose concentration (400 g/L) was used [243]. Bacteria of the genus *Bacillus* can synthesize a significant amount of levan. It was shown that in cultivation for 21 h in media supplemented with 20% sucrose, *B*. *subtilis* (natto) can produce up to 40–50 g/L of levan [244]. During batch and continuous cultivation in a bioreactor, the levan yield was 61 and 100 g/L, respectively [243]. Moreover, the *B*. *subtilis* strain CCT 7712 was able to synthesize up to 111.6 g/L of levan from 400 g/L sucrose media during 16 h cultivation [245].

*Paenibacillus polymyxa*, the type species of the genus *Paenibacillus*, is mainly isolated from plant-associated environments. For example, *P. polymyxa* 92, isolated from wheat roots, produced large amounts (38.4 g/L) of exopolysaccharide in a liquid nutrient medium containing 10% (*w*/*v*) sucrose and may serve as a basis for eco-friendly low-cost soil-remediation technologies owing to levan production exhibiting metal-binding ability, low viscosity in an aqueous solution, and plant-growth-regulating activity [246]. The *P. lautus* strain was isolated from Obsidian Hot Spring in Yellowstone National Park [247]. A new strain of *P. polymyxa* S3 with an antagonistic effect on 11 major fish pathogens was screened from the sediment of fishponds [248]. *P. antarcticus* IPAC21, the psychrotolerant bioemulsifier levan-producing strain, was isolated from the soil collected in Ipanema, King George Island, Antarctica [249]. Its genome was sequenced using Illumina Hi-seq, and a search for genes related to the production of bioemulsifiers and other metabolic pathways was carried out. The IPAC21 strain has a genome of 5,505,124 bp and a G + C content of 40.5%. Mamphogoro et al. (2022) reported the whole-genome sequence of *P. polymyxa* SRT9.1, a plant growth-promoting bacterium consisting of 6,754,470 bp and 7878 coding sequences, with an average G + C content of 45% [250]. Revin and Liyaskina et al. obtained a highly productive strain *Paenibacillus polymyxa* 2020, first isolated from wasp honeycombs capable of producing a greater amount of levan compared to the known *P*. *polymyxa* strains (Figure 3) [66]. The complete genome sequencing and methylome analysis of *P*. *polymyxa* 2020 were performed. The complete genome sequence of *P*. *polymyxa* 2020 is also available in GenBank with the accession numbers: CP049598-CP049599. The original sequence reads have been deposited at NCBI under SRA: SRR11236808; SRR11236809; SRR11236810; SRR11236811. Biosample: SAMN14247689. Bioinformatic analysis identified a putative levan synthetic operon. S*acC* and *sacB* genes were cloned; their products were identified as glycoside hydrolases and levansucrase. The highest levan yield of 68.0 g/L was obtained on a molasses medium with a total sugar concentration of 200 g/L. The highest yield of EPS in the sucrose medium was 53.78 g/L with 150 g/L sucrose at 96 h. Additionally, compared to the media with sucrose, in the molasses medium, the maximum polysaccharide production was achieved over a shorter time interval (48–72 h).

*Paenibacillus* spp. EPSs recently have attracted great attention due to their biotechnological potential in different industrial and wastewater treatment processes [251,252]. As of April 2023, according to the website List of Prokaryotic names with Standing in Nomenclature (LPSN), the genus *Paenibacillus* contains 374 species. The type of strain of *P*. *polymyxa* ATCC 842^T^ (=DSM 36^T^ = KCTC 3858^T^) has been deposited in a number of microbial collections. The bacterium was very often associated with the plant root microbiome, where it participates in bioprotection by the synthesis of antibiotics, phytohormones, hydrolytic enzymes, and EPS [256]. The genome sequences of 212 strains of *Paenibacillus* representing 82 different species are available [251]. The genome sizes range from 3.02 Mb for *P*. *darwinianus* Br isolated from Antarctic soil [253] to 8.82 Mb for *P*. *mucilaginosus* K02 implicated in silicate mineral weathering [254], and the number of genes varies from 3064 for *P*. *darwinianus* Br to 8478 *P*. *sophorae* S27. The insect pathogens *P*. *darwinianus*, *P*. *larvae,* and *P*. *popilliae* have smaller genomes from 4.51 and 3.83 Mb, respectively, which is likely to reflect their niche specialization. The GC content of *Paenibacillus* DNA ranges from 39 to 59% [255]. Complete genome sequencing and assembly has been done for strains such as *P*. *polymyxa* E681 [257], *P*. *polymyxa* SC2 [258], *P*. *polymyxa* ATCC 842T [259], *P*. *polymyxa* 2020 (isolated from wasp honeycombs) [66], *P*. *polymyxa* KF-1, soil isolated producer of antibiotics [260], and *Paenibacillus* sp. Y412MC10 [247], isolated from Obsidian Hot Spring in Yellowstone National Park, Montana, USA. Most of the deposited genome sequences are still in shotgun form. 

An EPS synthesizing potentially probiotic Gram-positive bacterial strain *Lactobacillus reuteri* was isolated from fish guts, and its EPS was structurally characterized. Based on molecular weight (MW) distribution, two groups of levan were found to be produced by the isolate FW2: one with a high MW (4.6 × 10^6^ Da) and the other having a much lower MW (1.2 × 10^4^ Da). The isolate yielded about 14 g/L levan under optimized culturing parameters [261]. The levan yield of the strain *Leuconostoc citreum* BD1707 reached 34.86 g/L with an MW of 2.320 × 10^7^ Da within 6 h cultivation [262].

Many Gram-negative acetic acid bacteria (AAB) of the *Alphaproteobacteria* class are also levan producers. *Acetobacteraceae* are known for their production of high-value homopolysaccharides such as levan [224,225,226]. *Gluconobacter japonicus* LMG 1417 is a potent levan-forming organism. A cell-free levan production based on the supernatant of the strain under study led to a final levan yield of 157.9 ± 7.6 g/L, and the amount of secreted levansucrase was more than doubled by plasmid-mediated homologous overproduction of LevS1417 in *G. japonicus* LMG 1417 [224]. *K. xylinus*, the producer of bacterial cellulose can also produce levan [227]. The good levan producer *Pseudomonas fluorescens* strain ES, with promising antioxidant and cytotoxic activity against different kinds of cancer cells, was isolated from soil in Egypt [231]. Possible levan producers were also identified among Gram-negative halophilic *Gammaproteobacteria* of the *Halomonas* genus [232] including *Halomonas smyrnensis* AAD6^T^ [233]. Their ability to grow in high concentrations of NaCl can be used to solve the problem of sterility in an industrial setting. The type of strain *H. smyrnensis* sp. nov. AAD6^T^ (=DSM 21644^T^ = JCM 15723^T^) was isolated in Turkey [233]. *Zymomonas mobilis* also produces levan. The strains ZAG-12 [228] and ATCC 31821 [229] were able to produce levan with approximate yields of 14.67 and 21.69 g/L, respectively. In continuous cultivation of *Z*. *mobilis* CCT4494, when immobilized in Ca-alginate gel, the amounts of levan were reported to be able to range from 18.84 up to 112.53 g/L depending on the incubation time [230]. Levan-type EPS is also produced by such *Gammaproteobacteria* as *Pantoea agglomerans* ZMR7. The maximum levan production (28.4 g/L) was achieved when sucrose and ammonium chloride were used as carbon and nitrogen sources, respectively, at 35 °C and an initial pH of 8.0 [263]. *Gammaproteobacteria Erwinia amylovora*, the causative agent of fire blight, is also a levan producer [234]. Several recombinant *E. coli* and yeasts were developed to study the biochemistry of levan synthesis by cloning and expression of levansucrase genes from *L. mesenteroides* and *B. amyloliquefaciens* [264,265,266,267,268,269].

Levan is synthesized by levansucrase (E.C 2.4.1.10), a fructosyltransferase belonging to the family of glycoside hydrolases [221]. Levansucrase binds to a substrate, such as sucrose, and adds fructose molecules to a growing fructose chain [221]. In Gram-negative bacteria, levansucrase is secreted through a single peptide pathway, which promotes sucrose hydrolysis and levan formation through transfructosylation activity [270]. A high concentration of substrate has been found to inhibit levansucrases in Gram-negative bacteria since the interaction of enzyme-substrate occurs in the periplasmic space resulting in the accumulation of enzyme and product, while in Gram-positive bacteria, this is not observed due to the presence of peptidoglycan wall [222]. In terms of its biological role, levan is involved in biofilm formation in some bacteria [271,272] and also contributes to the fitness and virulence of plant pathogens [221,273]. Further, in soil-resident bacteria, levan promotes salt tolerance and desiccation, as well as the formation of cell aggregates on abiotic surfaces [272]. In addition, levan can serve as a source of reserve substance under starvation conditions [221]. Moreover, levan has been suggested to promote the colonization of bacteria in the gut [274] and to act as a prebiotic in vitro [275,276]. 

Levan is an industrially important, functional biopolymer widely applied in food, biomedicine, cosmetic, and pharmaceutical fields owing to its safety and biocompatibility. In the food industry, levan can be used as a gelling agent, as well as a food supplement with prebiotic properties [220,277,278]. Moreover, microbial levan can be a good source of pure fructose production. The bio-degradable film obtained with levan has a high potential to be used in different areas, especially in food packaging [279]. In the biomedical sectors, levan has many applications due to its biocompatibility and antibacterial [215,218], anticancer [215,216,217], antioxidant [218], anti-inflammatory, immunomodulatory [219], and prebiotic activities [220]. Levan was described as an effective therapeutic agent in some human conditions, such as cancer, heart disease, and diabetes. Levan treatment is a promising therapeutic strategy for neuroblastoma and osteosarcoma cells [216,217]. Levan is a promising biopolymer to develop nanomaterials. Due to its amphiphilic properties, the main application of levan so far is the production of nanoparticles. For example, vancomycin was encapsulated in levan nanoparticles of 200–600 nm [280]. Nanoparticle complexes with levan could be proposed as potent drug delivery vehicles for cancer drugs, as well as other drugs in prospective studies [281,282]. Kim et al., encapsulated a marker (indocyanine green) in levan nanoparticles (100–150 nm) for tumor imaging [283]. Tabernero et al., attached 5-fluorouracil on levan nanoparticles for colorectal cancer treatment [282]. Silver and gold nanoparticles were previously coated with this polymer to be used as a catalyst [284] or a bactericidal system [222]. There were obtained from levan-based nanoparticles with resveratrol, which can be used in drug delivery systems, wound healing, and tissue engineering [285]. Levan nanoparticles can also be used as drug carriers for peptides and proteins [280]. In addition, it can provide biocompatible surfaces, especially in tissue engineering [286,287,288]. Sulfated levan can be considered a promising material for cardiac tissue engineering applications [287]. It is related to its excellent biocompatibility and anticoagulant activity. Gomes et al., developed self-adhesive free-standing multilayer films from sulfated levan combined with alginate and chitosan [286]. The presence of sulfated levan significantly improved the mechanical strength and adhesiveness of the constructed adhesive films. The multilayer films were cytocompatible and myoconductive, which was assessed through in vitro testing on a myoblast cell line, C2C12. Levan nanocomposite films have the potential to be used in industrial and medical fields [289]. Levan also has appropriate properties for cosmetic applications, for example, in safe and functional body wash cosmetics production [290,291]. Levan has a film-forming potential and high adsorption properties. Therefore, it can be used as an adsorbent for heavy metals in industrial wastewater treatment [246]. Furthermore, levan has higher adhesive properties than carboxymethyl cellulose and can be used for wood bonding. Its availability as a biological binder to obtain wood biocomposite materials was shown [292]. The resulting composites were demonstrated to have a more uniform structure and good strength characteristics. Levan is a promising material for obtaining biocomposites. It can be processed with other polysaccharides or functionalized to produce fibers or gels. For example, a sulfated derivative of levan was processed with polycaprolactone (PCL) and polyethyleneoxide (PEO) using electrospinning technology. The obtained microfibers (1–5 microns in diameter) improved blood clotting by adding levan. The work indicated the potential of using levan in blends to produce anti-thrombogenic compounds [288].

## 4. Xanthan-Producing Bacteria 

Xanthan is one of the industrially most relevant bacterial polysaccharides. Compared to other microbial polysaccharides, its price is competitive, and, therefore, its production is cost-effective [293]. Xanthan is an important industrial biopolymer, which, due to its beneficial properties, has found application in food, oil, pharmaceutical, mining, textile, and other industries [294,295,296,297]. It is environmentally friendly, non-toxic, and has the following unique properties: stability at high temperature and salinity, pseudoplasticity, highly viscous even at very low concentrations, chemical stability for oil well conditions, biodegradability, reasonable cost, and environmental friendliness [298]. The global xanthan gum market was valued at ~USD 1 billion in 2019 and is expected to reach ~USD 1.5 billion in 2027 [4]. Xanthan is an anionic heteropolysaccharide consisting of a cellulose-like backbone of β-1,4-linked glucose units, substituted alternately with a trisaccharide side chain, which is composed of two mannose units separated by a glucuronic acid, where the internal mannose is mostly O- acetylated and the terminal mannose may be substituted by a pyruvic acid residue. Due to the presence of glucuronic and pyruvic acid in the side chain, xanthan represents a highly charged polysaccharide with a very rigid polymer backbone. It has a high molecular weight of about 2 × 10^6^ to 2 × 10^7^ g mol^−1^.

*X*anthan is produced by different *Xanthomonas* sp. The genus *Xanthomonas* belongs to the class *Gammaproteobacteria* of the phylum *Proteobacteria*. As of April 2023, according to the website List of Prokaryotic names with Standing in Nomenclature (LPSN), the genus *Xanthomonas* contains 53 species, some of which cause economically important diseases in more than 400 host plants [33]. *Xanthomonas* sp. has two features: the formation of exopolysaccharide xanthan and the formation of specific membrane-bound pigments—xanthomonadins that provide the mucoid and yellow color of the colonies. *Xanthomonas* cells are usually rod-shaped single-cell ones with a single polar flagellum. The type of strain is *Xanthomonas campestris* ATCC 33913^T^ (=CFBP 2350^T^ = CIP 100069^T^ = DSM 3586^T^ = ICMP 13^T^ = LMG 568^T^ = NCPPB 528^T^). The *Xanthomonas* spp. differentiate further into pathovars depending on the host plant [34,299]. For example, *Xanthomonas campestris* pv. *campestris* is the causal agent of black rot disease affecting many crop plants from the *Brassicaceae* family [300]. The bacterium *Xanthomonas oryzae* pv. *oryzae* causes rice bacterial leaf blight, one of the most destructive rice diseases [301]. Bacteria deploy a large arsenal of virulence factors to successfully infect the host plant. In particular, xanthan protects bacterial cells against environmental stresses and supports biofilm formation [299]. QS coordinates bacterial behavior including biofilm dispersal and is required for disease [299]. Currently, Feng et al. (2023) provided a comprehensive review of the recent advances in diffusible signal factor (DSF)-mediated QS in *Xanthomonas* and reported the inhibitors that are proposed as bactericide candidates to target the RpfF enzyme and control plant bacterial diseases [302]. The QS in *Xanthomonas* is associated with rpf (regulation of pathogenicity factor) genes, among which, the main genes *rpfF*, *rpfB*, *rpfC*, and *rpfG* encode proteins involved in DSF synthesis, turnover, sensing, and transduction, respectively [302]. The *Xanthomonas* genus includes species such as *X. campestris*, *X. arboricola*, *X. axonopodis*, *X. fragaria*, *X. gummisudans*, *X. juglandis*, *X. phaseoli*, *X.vasculorium*, and others. The *Xanthomonas campestris* bacterium is used in industrial production.

The complete genome sequences of many *Xanthomonas* strains have been determined to date. Liyanapathiranage et al. analyzed 1740 complete genome sequences belonging to 39 *Xanthomonas* spp. retrieved from the National Center for Biotechnology Information (NCBI), a large proportion of them being from the species *X. oryzae* (396 genomes), *X. citri* (195 genomes), *X. phaseoli* (97 genomes), *X. perforans* (151 genomes), and *X. arboricola* (136 genomes) [303]. The authors studied the genetic organization and patterns of evolution of several clusters of the type VI secretion system (T6SS) in order to understand the contribution of T6SS toward the ecology and evolution of *Xanthomonas* spp. Recently, Cuesta-Morrondo et al. (2022) reported that the resulting *X. arboricola* pv. *pruni* IVIA 2626.1 genome comprised two circular contigs, a 5.11 Mb chromosome and a 41.10 kb plasmid, while CITA 33 was composed of a 5.09 Mb chromosome and 41.11 kb plasmid [304]. In addition, Bellenot et al. (2022) reported on draft genome sequences of 17 strains representing eight of nine known races of the pathogen *Xanthomonas campestris* pv. *campestris* causing black rot disease on *Brassicaceae* crops [305]. Revin et al. obtained a highly efficient xanthan-producing strain, *X.campestris* M 28, which produced up to 28 g/L of the polysaccharide on a molasses medium (Figure 4). Whole-genome sequencing of the strain was performed using the Illumina method and nanopore sequencing. The genome of *X. campestris* M 28 contained one chromosome of 5 102 828 nucleotides with an average G + C content of 65.03% [306]. The genomes of the strains *X. campestris* NRRL B-1459 (ATCC 13951) [307], *X. campestris pv. campestris* B 100 [308], *X. campestris* JX [309], *X. campestris* pv. *campestris* WHRI 3811 [310], and others, also have been completely described. The size of *X. campestris* chromosomes was shown to range from 4.8 to 5.1 Mb [307,308,309]. The content of GC bases in the *X. campestris* chromosome is 63.7–65.3% [307,308,309]. A number of *X. campestris* strains were established to have plasmids of various lengths responsible for resistance to antibiotics, metals, etc. *Xanthomonas* genomes comprise different mobile genetic elements, such as transposons, insertion sequence, plasmids, and genomic islands associated with virulence factors, genetic variations, and genome structure [311]. 

Xanthan is synthesized through a Wzy-dependent pathway, which comprises several steps including the synthesis of exopolysaccharide precursors (nucleotide sugars GDP-mannose, UDP-glucose, and UDP-glucuronic acid), repeat-unit assembly on a lipid carrier located at the cytoplasmic membrane, membrane translocation to the periplasmic face, polymerization by a block-transfer mechanism involving Wzy polymerase, and export [312]. The building of pentasaccharide units is under the control of the gum cluster comprising 12 genes involved in the pentasaccharide repeating unit assembly, its decoration with substituents, polymerization, translocation, and secretion [312]. However, the key factors for the xanthan’s precursors’ biosynthesis are the genes *xanA* and *xanB*, which are not included in the gum cluster. The *xanA* and *xanB* genes are involved in UDP-glucose and GDP-mannose biosynthesis.

Like other EPSs, xanthan yield and quality can be modified using different bacterial strains and fermentation environments (e.g., carbon source, temperature, pH, mixing speed, inoculum volume, and airflow rate). The carbon source is an essential factor in microbial xanthan fermentation, which acts as an energy source and is used in EPS synthesis. Media with complex compositions are most often used to obtain xanthan; these mainly include 2–4% glucose and sucrose and 0.05–0.1% nitrogen sources, such as yeast extract, peptone, and ammonium nitrate [297], and have quite a high cost. Media including various industrial and agricultural wastes have been proposed to reduce the cost of xanthan [306,313]. This also solves the environmental problems of waste disposal, which negatively affect the environment condition. The immobilization of bacteria cells (e.g., *X. campestris* and *X. pelargonii*) on calcium alginate-based beads showed higher xanthan yield compared to free cells, irrespective of the carbon source [314].

Xanthan is a biopolymer that has found application in food, petroleum, pharmaceutical, mining, textile, cosmetics, and other industries [3,294,295,296,297,298,315,316,317,318,319]. This polymer is environmentally friendly and non-toxic and, therefore, used in the food industry for the last 50 years as a stabilizer and thickener. Xanthan is a promising material for medical applications since it is a biocompatible polymer and not cytotoxic. Recently, it attracted the particular attention of researchers in connection with the prospects for its use in tissue engineering, drug delivery, and obtaining biocomposites with regenerative and antibacterial properties [4,316,317]. Recently, Barbosa et al. (2023) produced cell-friendly chitosan-xanthan composite membranes incorporating hydroxyapatite to be used in guided tissue and bone regeneration, in particular, for periodontal tissue regeneration [316]. The review by Jadav et al. (2023) described xanthan applications of xanthan in delivering various therapeutic agents such as drugs, genetic materials, proteins, and peptides [317]. Xanthan can be used as a new green-based material to produce superabsorbents and for the remediation of contaminated waters. Sorze et al. (2023) developed novel biodegradable hydrogel composites of xanthan gum and cellulose fibers to be used both as soil conditioners and topsoil covers to promote plant growth and forest protection [318]. The rheological, morphological, and water absorption properties of produced hydrogels were comprehensively investigated. Specifically, the moisture absorption capability of these hydrogels was above 1000%, even after multiple dewatering/rehydration cycles. A recent review by Balíková et al. (2022) highlighted xanthan prospects as green adsorbents for water decontamination [312]. A number of scientists have shown the functional groups of xanthan to be able to bind heavy metals from aqueous solutions and effectively remove them. For example, Wang et al. showed the superior adsorption capacity of xanthan produced by *X. campestris* CCTCC M2015714 to detoxify lead (II), cadmium (II), and copper (II) polluted waters with an efficiency of 50% in an hour [319]. The review by Dzionek et al. (2022) summarized cross-linking methods that could potentially be used to reduce their toxicity to living cells and demonstrated xanthan availability for whole-cell immobilization with the prospect of using it in bioremediation [320]. The polysaccharide xanthan is considered to be the most commonly used polymer for enhanced oil recovery [298]. It is relatively more stable under harsh conditions and has the necessary rheological properties. Recently, xanthan has received attention for its application in 3D printing technology. At low concentrations, xanthan has shear thinning capacity and required viscosity due to which it can function as a rheological modifier, thus improving 3D printing potential [321,322].

## 5. Conclusions and Future Perspective

The present review summarizes the current progress in research on bacterial EPS, mainly over the past 5 years, including their properties, biological functions, and promising applications in various fields of science, industry, medicine, and technology. Such EPS as BC, levan, and xanthan are described in detail. The information on the systematic position, sources of isolation and properties of EPS-producing bacteria is presented. Bacteria are characterized by metabolic flexibility and a variety of physiological and biochemical properties. Therefore, using the techniques optimizing cultivation conditions, genetic, and metabolic engineering, it is possible to modulate the yield as well as the structural and functional properties of bacterial EPS. At present, the biochemical and molecular basis of the biosynthesis of BC, xanthan, and levan are well studied. The genomes of a large number of bacteria have been fully sequenced. All this is the basis for the metabolic regulation of EPS biosynthesis and the creation of resistant genetically engineered strains that have not yet been obtained for industrial production. The yield of the most known bacterial EPS is still low enough for industrial production. Therefore, the isolation of new producers and the production of highly productive strains by genetic and metabolic engineering are very relevant. The investigation of unique polysaccharides from marine bacteria and extremophiles is now the focus of scientific research. They have considerable potential in bioremediation, water purification from heavy metal, and marine oil pollution and also in pharmaceutical and biomedical fields as antioxidants, antifreeze, anti-cancer, anti-inflammatory, antibacterial agents, etc. Although these EPS are well used in many areas, there are relatively few studies on them compared to land microorganisms, especially the studies on EPS formed in extreme marine environments, partially due to the difficulty of isolation and culture conditions. Therefore, new and updated technical strategies are needed to isolate producers and analyze novel marine microbial EPS. Since the study of bacterial polysaccharides is partly hindered by the incapability of culturing methods, metagenome-based techniques should be used to search for and study new bacterial polysaccharides. In recent years, valuable information has emerged on QS mechanisms in EPS producers. This area requires further study in terms of application for more efficient EPS production. Due to the high cost of EPS production, a large number of publications have recently suggested approaches to solve the problem, in particular, the use of various agricultural and industrial wastes. At the same time, their disposal burning problem is also being solved. Another promising area is biocatalytic technologies and cell-free synthesis to obtain EPS more efficiently. Bacterial EPS are characterized by the presence of a large number of functional groups (hydroxyl, carboxyl, carbonyl, acetate, etc.), which enables them to modify their molecules in order to give them new valuable properties, such as antimicrobial activity, etc. Therefore, a large number of EPS-based biocomposite materials have been obtained. The already developed methodological approaches and the accumulated data on their modification will enable the creation of an even greater number of different functional and structural materials of a new generation with a wide range of applications in the future.

## Figures and Tables

**Figure 1 microorganisms-11-01541-f001:**
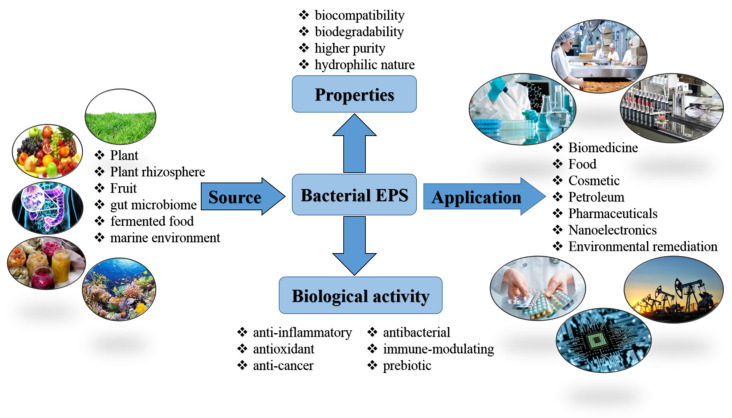
Bacterial EPS properties, biological activity, application, and sources.

**Figure 2 microorganisms-11-01541-f002:**
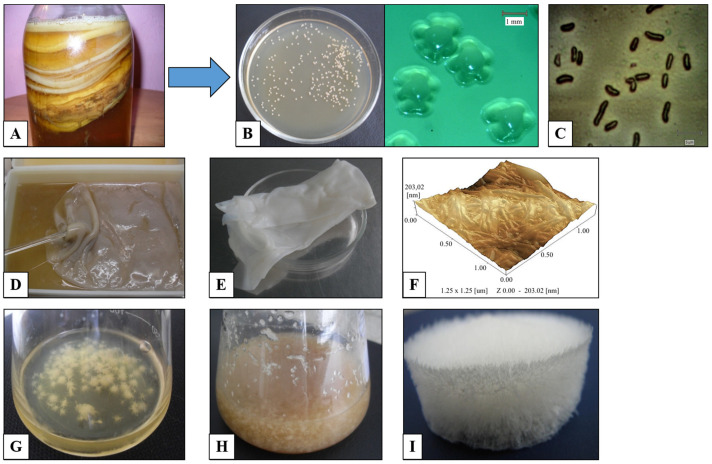
*K. sucrofermentans* B-11267 isolated from the kombucha community (**A**). The colony morphology (**B**). Cell morphology (scale bar: 5 μm) (**C**). Gel film obtained in static conditions (**D**). Gel film after purification (**E**). AFM (atomic force microscopy) image of BC (**F**). BC agglomerates of various shapes formed in agitated culture conditions (**G**,**H**). BC aerogel (**I**).

**Figure 3 microorganisms-11-01541-f003:**
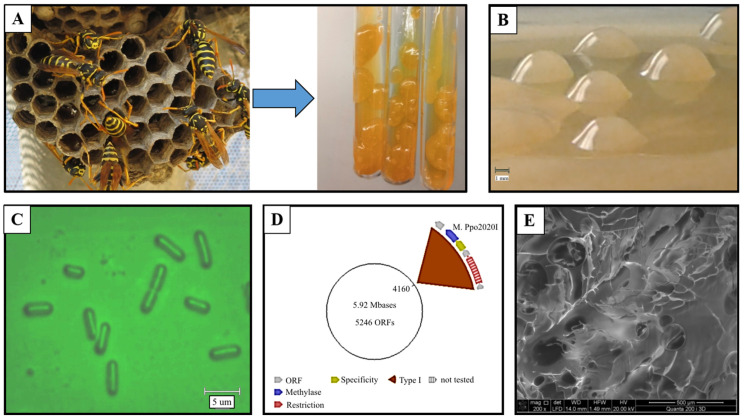
*Paenibacillus polymyxa* 2020, first isolated from wasp honeycombs (**A**). The colony morphology (**B**). Cell morphology (**C**). Chromosome map (**D**). SEM image of EPS (**E**). Adapted with permission from Ref. [66].

**Figure 4 microorganisms-11-01541-f004:**
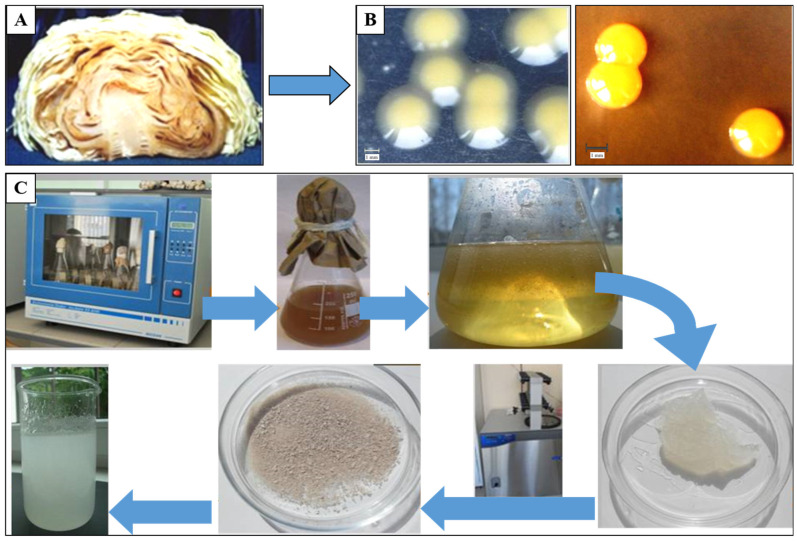
*X.campestris* M 28 isolated from cabbage (**A**). The colony morphology (**B**). Xanthan production (**C**).

**Table 1 microorganisms-11-01541-t001:** General properties of the type strains of species of the genus *Komagataeibacter*.

No	Species	Sources of Isolation	Number of Bases (bp)	DNA G + C Content (mol%)	Ref.
1	*K. cocois* WE7^T^ (CGMCC 1.15338^T^; JCM 31140^T^)	Contaminated coconut milk, China	3,406,946	62.7	[92]
2	*K. diospyri* MSKU 9^T^ (NBRC 113802^T^; TBRC 9844^T^)	Persimmon, Thailand	3,762,373	60.4	[93]
3	*K. europaeus* LMG 18890^T^ (ATCC 51845^T^; DES 11^T^; DSM 6160^T^)	Submerged culture vinegar generator, Germany	4,227,398	61.3	[90]
4	*K. hansenii* JCM 7643^T^ (DSM 5602^T^; ATCC 35959^T^; BCC 6318 ^T^)	Vinegar, Israel	3,710,965	59.3	[94]
5	*K. intermedius* TF2^T^ (DSM 11804^T^; BCC 36457^T^; JCM 16936^T^; LMG 18909^T^; BCC 36447^T^; BCRC 17055^T^; CIP 105780^T^)	Kombucha beverage, Switzerland	3,883,532	61.6	[95]
6	*K. kakiaceti* JCM 25156^T^ (DSM 24098^T^; G5-1^T^; LMG 26206^T^; NRIC 798^T^; BCRC 80743^T^)	Kaki vinegar, Japan	3,133,102	62.1	[96]
7	*K. kombuchae* LMG 23726^T^ (MTCC 6913^T^; RG3^T^)		3,483,869	59.6	[94]
8	*K. maltaceti* LMG 1529^T^ (IFO 14815^T^; NBRC 14815^T^; NCIB 8752^T^; NCIMB 8752^T^)	Malt vinegar brewery acetifier	3,638,012	63.2	[96,97]
9	*K. medellinensis* NBRC 3288^T^ (IFO 3288^T^); Kondo 51^T^; LMG 1693^T^)	Vinegar, Japan	3,136,818	60.9	[96,98]
10	*K. melaceti* AV382^T^ (ZIM B1054^T^; LMG 31303^T^; CCM 8958^T^)	Apple vinegar	3,629,663	59.14%	[99]
11	*K. melomenusus* AV436^T^ (ZIM B1056^T^; LMG 31304^T^; CCM 8959^T^)	Apple cider Vinegar, Kopivnik, Slovenia			[99]
12	*K. nataicola* LMG 1536^T^ (BCC 36443^T^; JCM 25120^T^; NRIC 616^T^)	Nata de coco, Philippines	3,672,972	61.5	[90,98]
13	*K. oboediens* LMG 18849^T^ (BCC 36445^T^; CIP 105763^T^; DSM 11826^T^ JCM 16937^T^; NCIMB 13557^T^; LTH 2460^T^;BCRC 17057^T^)	Submerged red wine vinegar, Germany	3,777,265	61.4	[90]
14	*K. pomaceti* T5K1^T^ (CCM 8723^T^; LMG 30150^T^; ZIM B1029^T^)	Apple cider vinegar, Slovenia	3,449,370	62.5	[94]
15	*K. rhaeticus* LMG 22126^T^ (DSM 16663^T^, BCC 36452^T^; JCM 17122^T^; DST GL02^T^;CIP 109761^T^)	Apple juice, South Tyrol region, Italy	3,465,200	63.5	[90,98]
16	*K. saccharivorans* LMG 1582^T^ (BCC 36444^T^; JCM 25121^T^; NRIC 614; CIP 109786^T^; CECT 7869^T^; NCCB 29003^T^;NRIC 0614^T^)	Beet juice, Germany	3,350,941	61.6	[90,98]
17	*K. sucrofermentans* LMG 18788^T^ (DSM 5973^T^; BCC 7227^T^; JCM 9730^T^; BPR 2001^T^; ATCC 700178^T^; BCRC 80162^T^; CECT 7291^T^;CIP 106078^T^)	Black cherry, Tokyo, Japan	3,363,922	62.3	[90]
18	*K. swingsii* LMG 22125^T^ (DSM 16373^T^; BCC 36451^T^; JCM 17123; DST GL01^T^; CIP 109760^T^)	Apple juice, South Tyrol region, Italy	3,732,982	62.4	[90]
19	*K. xylinus* LMG 1515^T^ (ATCC 23767^T^; DSM 6513^T^; BCC 7226^T^; JCM 7644^T^; NBRC 15237^T^; NCIMB 11664^T^; BCRC 2952^T^; CCM 3611^T^; CCUG 37299^T^; CECT 7351^T^; CIP 103107^T^; IFO 15237^T^; NCTC 4112^T^; VTT E-97831^T^)	Mountains ash berries	3,660,954	66.2	[90,98]

## Data Availability

Sequence data are available from GenBank, NCBI.

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
