# Peer review of "Exopolysaccharides Producing Bacteria: A Review"

_microorganisms, 2023, doi:10.3390/microorganisms11061541_

Round 1

Reviewer 1 Report

Comments to the Author

This review article summarizes the current progress in research on bacterial EPS including their properties, biological functions, and promising applications in various fields. This review article interesting and worthy of publication after consideration of the following:

 “Microorganisms 2020, 8, x FOR PEER REVIEW” should be “Microorganisms 2023, 8, x FOR PEER REVIEW”

line numbering is essential

page 2& 18:  font size should be unified through the manuscript

various fields including medicine, food, and environmental protection, etc. [18, 19].       

stress, phagocytosis, and chemical agents (antibiotics, heavy metals, oxidants) [2].

The abbreviation (QS) is for“quorum sensing” (page 9) NOT quorum-sensitive as in page 4& page 19  so please adjust this point through the manuscript.

In page 16 “. Quorum sensing (QS) coordinates bacterial behavior”:  please write the abbreviation (QS) as it was mentioned before in page 9

 page 3: “exopolysaccharides formed by it, the first of exopolysaccharides consists of mannose,” & “to inhibit biofilm formation by targeting exopolysaccharides [32]. EPS play an important” replace “exopolysaccharides” with “EPSs”

page 12 “A new exopolysaccharide producing Gram-positive bacterium”: replace “exopolysaccharides” with “EPSs”

Page 5 “bacteria species of Komagataeibacter genus such as K. xylinus and K. hansenii.”  & page 8” K. intermedius [134], G. swingsii [135], K. rhaeticus DSM 16663T [131], G. rhaeticus”:

 Bacterial species should be italicized.

In Table 1 footnote please clarify the underlined strains inside the table.

page 11” uniquephysicochemical properties” should be replaced with “unique physicochemical properties”

page 12 “genome sequence of P. polymyxa2020”: take a space

Comments to the Author

This review article summarizes the current progress in research on bacterial EPS including their properties, biological functions, and promising applications in various fields. This review article interesting and worthy of publication after consideration of the following:

 “Microorganisms 2020, 8, x FOR PEER REVIEW” should be “Microorganisms 2023, 8, x FOR PEER REVIEW”

line numbering is essential

page 2& 18:  font size should be unified through the manuscript

various fields including medicine, food, and environmental protection, etc. [18, 19].       

stress, phagocytosis, and chemical agents (antibiotics, heavy metals, oxidants) [2].

The abbreviation (QS) is for“quorum sensing” (page 9) NOT quorum-sensitive as in page 4& page 19  so please adjust this point through the manuscript.

In page 16 “. Quorum sensing (QS) coordinates bacterial behavior”:  please write the abbreviation (QS) as it was mentioned before in page 9

 page 3: “exopolysaccharides formed by it, the first of exopolysaccharides consists of mannose,” & “to inhibit biofilm formation by targeting exopolysaccharides [32]. EPS play an important” replace “exopolysaccharides” with “EPSs”

page 12 “A new exopolysaccharide producing Gram-positive bacterium”: replace “exopolysaccharides” with “EPSs”

Page 5 “bacteria species of Komagataeibacter genus such as K. xylinus and K. hansenii.”  & page 8” K. intermedius [134], G. swingsii [135], K. rhaeticus DSM 16663T [131], G. rhaeticus”:

 Bacterial species should be italicized.

In Table 1 footnote please clarify the underlined strains inside the table.

page 11” uniquephysicochemical properties” should be replaced with “unique physicochemical properties”

page 12 “genome sequence of P. polymyxa2020”: take a space

Author Response

Thank you very much for all of your detailed comments and suggestions. We found them quite useful as we approached our revision. The authors completely agree with reviewers comment. We would like to thank the reviewers for their work. We have been attentive to their comments and made corrections to our article. We would also like to thank the Editor for the opportunity to publish our work in the journal Microorganisms.

 (Reviewer 1

Thank you very much for all of your detailed comments and suggestions. We found them quite useful as we approached our revision. The authors completely agree with reviewers comment.

  1. “Microorganisms 2020, 8, x FOR PEER REVIEW” should be “Microorganisms 2023, 8, x FOR PEER REVIEW”.

RESPONSE: Thank you very much for your correction. The text was changed.

  1. Line numbering is essential

RESPONSE: The lines are numbered.

  1. Page 2& 18:  font size should be unified through the manuscript.

RESPONSE: Font size was unified through the manuscript.

  1. The abbreviation (QS) is for“quorum sensing” (page 9) NOT quorum-sensitive as in page 4& page 19  so please adjust this point through the manuscript.

RESPONSE: This point was corrected through the manuscript.

  1. In page 16 “. Quorum sensing (QS) coordinates bacterial behavior”:  please write the abbreviation (QS) as it was mentioned before in page 9

RESPONSE: Thank you very much for your correction. The text was changed.

  1. Page 3: “exopolysaccharides formed by it, the first of exopolysaccharides consists of mannose,” & “to inhibit biofilm formation by targeting exopolysaccharides [32]. EPS play an important” replace “exopolysaccharides” with “EPSs”

RESPONSE: The text was changed.

  1. Page 12 “A new exopolysaccharide producing Gram-positive bacterium”: replace “exopolysaccharides” with “EPSs”

RESPONSE: The text was changed.

  1. Page 5 “bacteria species of Komagataeibacter genus such as K. xylinus and K. hansenii.”  & page 8” K. intermedius [134], G. swingsii [135], K. rhaeticus DSM 16663T [131], G. rhaeticus”:  Bacterial species should be italicized.

RESPONSE: Bacterial species names corrected to italics.

  1. In Table 1 footnote please clarify the underlined strains inside the table.

RESPONSE: The underlined strains inside the table is a mistake. Text was corrected.

  1. page 11” uniquephysicochemical properties” should be replaced with “unique physicochemical properties”

RESPONSE: The text was corrected.

  1. Page 12 “genome sequence of P. polymyxa2020”: take a space

RESPONSE: The text was corrected.

Reviewer 2 Report

The authors aimed to summarize the current research progress on bacterial bacterial exopolysaccharides with special attention paid to the bacterial cellulose, xanthan and levan producing bacteria.

Overall, the manuscript is good and can be accepted in its current form after a minor revision.

The authors should review the name of bacterial species and improve the figures.

Author Response

Thank you very much for all of your detailed comments and suggestions. We found them quite useful as we approached our revision. The authors completely agree with reviewers comment. We would like to thank the reviewers for their work. We have been attentive to their comments and made corrections to our article. We would also like to thank the Editor for the opportunity to publish our work in the journal Microorganisms.

Thank you very much for all of your comments and suggestions. We found them quite useful as we approached our revision. The authors completely agree with reviewers comment.

  1. The authors should review the name of bacterial species and improve the figures.

RESPONSE: Thank you very much for your recommendation. Manuscript corrections have been made. Bacterial species names corrected to italics.

Reviewer 3 Report

The review has shed a deep insight on significance of bacterial exopolysaccharides (EPSs). The authors have done a meticulous literature review for this goal.

Major concerns:

1.       Lack of novelty is the biggest concern.

2.       More emphasis is given on literature for bacteria producing the EPSs. This is shifting the focus of the review from EPS to EPS producing bacteria. Information related to isolation, GC content and sequencing data for bacterial identification is not adding much value to the aim of this review.

 Minor editing of English language required

Author Response

Thank you very much for all of your comments and suggestions. We found them quite useful as we approached our revision. The authors completely agree with reviewers comment.

  1. Lack of novelty is the biggest concern.

RESPONSE: Over the past 2-3 years, a large number of innovative works have appeared in the field of studying such exopolysaccharides as bacterial cellulose (BC), levan and xanthan. The review provides information on whole genome sequencing of the novel BC, levan, and xanthan producing bacteria, which can be the basis for the metabolic regulation of EPS biosynthesis and the creation of highly productive strains. In addition, the review raises a novel problem of QS mechanisms in the producers of BC, levan and xanthan. Furthermore the review represented the novel applications of BC, xanthan and levan. We hope that this review will help researchers in the field of bacterial EPS as well as provide an important basis for the highly efficient production of BC, xanthan and levan and the development of new functional materials for a wide range of applications.

  1. More emphasis is given on literature for bacteria producing the EPSs. This is shifting the focus of the review from EPS to EPS producing bacteria. Information related to isolation, GC content and sequencing data for bacterial identification is not adding much value to the aim of this review.

RESPONSE: The aim of this review was to summarize the current research progress on bacterial EPSs with special attention paid to the bacterial cellulose, xanthan and levan producing bacteria. Information related to isolation and sequencing data is important for the obtaining highly productive strains and for the highly efficient production of these EPS.

  1. Pages 2, 3, 7, 9, 12, 14, 16-18 contain text in a different font size, please check.

RESPONSE: Font size was unified through the manuscript.

Reviewer 4 Report

This review is a good effort to describing EPS producing bacteria with a special focus on bacterial cellulose, levan and xanthan. Although the review is very comprehensive (it covers 341 literary sources), it is well written and easy to read. It certainly deserves publication in the journal Microorganisms after comments have been addressed: 

1. The title should be changed to a more appropriate one that reflects the particular focus of this work on cellulose, levan and xanthan producing bacteria.

2. The aim of the work should be better justified, what is the specific reason for choosing cellulose, levan and xanthan, and not other EPSs? 

2. Pages 2, 3, 7, 9, 12, 14, 16-18 contain text in a different font size, please check.

3. There are a lot of mistakes in punctuation and grey shadow (I guess these sentences were directly copied from other places), reflecting that the manuscript has not been well modified by the coauthors.

4. Please avoid very short paragraphs.

5.  Pages 5, 8 and 16 contan taxa names, some of which should be in italics.  Italics are used for bacterial taxa at the level of family and below. Please correct here and elsewhere in the text. Gene names should also be in italics (please see page 17)

6. Figure 2: please define AFM in the caption. Please check the correct spelling of the word airogel.

7. Page 8: "..the maintenance of aerobic conditions can be the reason for their formation [146]." not clear, please rephrase.

8. Page 8: "foreign substances" this expression is not used for bacteria, please rephrase

9. Page 8: "A biofilm gives bacteria the ability to horizontal gene transfer providing resistance.." please check if the meaning is correct

10. Page 11: "uniquephysicochemical properties" please check

11. Page 15: "rearded" please check this word

12. Page 15:  "Sulfated levan promising for future use in cardiac tissue engineering." please check English

13. Page 15: "with PCL and PEO"  Please define

14. Page 16: "Liyanapathiranage et al. analyzed 1,740 complete genome sequences belonging to 39 Xanthomonas spp. retrieved from the National Center for Biotechnology Information (NCBI), a large proportion of them being from the species: X. oryzae (396 genomes), X. citri (195 genomes), X. phaseoli (97 genomes), X. perforans (151 genomes), and X. arboricola (136 genomes) [320]." What were the main findings of this work?

The use of English is generally acceptable, minor corrections are recommended.

Author Response

Thank you very much for all of your detailed comments and suggestions. We found them quite useful as we approached our revision.

  1. 1. The title should be changed to a more appropriate one that reflects the particular focus of this work on cellulose, levan and xanthan producing bacteria.

RESPONSE: Thank you very much for your recommendation. The introduction provides general information about bacterial exopolysaccharides and it is noted that the review provides the current research progress on bacterial EPSs with special attention paid to the bacterial cellulose, xanthan and levan producing bacteria.

  1. The aim of the work should be better justified, what is the specific reason for choosing cellulose, levan and xanthan, and not other EPSs? 

RESPONSE: Thank you very much for your recommendation. These polysaccharides are of great industrial importance and have prospects for the production of novel functional materials for a wide range of applications. Currently, the biochemical and molecular basis of the biosynthesis of BC, xanthan, and levan are well studied. Recently, the genomes of a large number of bacteria have been fully sequenced. All this is the basis for the metabolic regulation of EPS biosynthesis and the creation of highly productive strains and the highly efficient production of BC, xanthan and levan.

  1. Pages 2, 3, 7, 9, 12, 14, 16-18 contain text in a different font size, please check.

RESPONSE: Thank you very much for your correction. Font size was unified through the manuscript.

  1. There are a lot of mistakes in punctuation and grey shadow (I guess these sentences were directly copied from other places), reflecting that the manuscript has not been well modified by the coauthors.

RESPONSE: Thank you very much for your correction. The text was changed.

  1. Please avoid very short paragraphs.

RESPONSE: Thank you very much for your recommendation. Very short paragraphs have been merged.

  1. Pages 5, 8 and 16 contain taxa names, some of which should be in italics.  Italics are used for bacterial taxa at the level of family and below. Please correct here and elsewhere in the text. Gene names should also be in italics (please see page 17)

RESPONSE: Thank you very much for your correction. Taxa names and gene names corrected to italics.

  1. Figure 2: please define AFM in the caption. Please check the correct spelling of the word airogel.

RESPONSE: The text was corrected.

  1. Page 8: "..the maintenance of aerobic conditions can be the reason for their formation [146]." not clear, please rephrase.

RESPONSE: The text was changed as follows: «For example, BC film on the medium surface ensures maintenance of an aerobic environment».

  1. Page 8: "foreign substances" this expression is not used for bacteria, please rephrase

RESPONSE: The expression "foreign substances" was rephrased to «external agents».

  1. Page 8: "A biofilm gives bacteria the ability to horizontal gene transfer providing resistance.." please check if the meaning is correct

RESPONSE: The text was changed as follows: «A biofilm gives bacteria the ability to horizontal gene transfer providing antibiotic resistance…»

  1. Page 11: "uniquephysicochemical properties" please check

RESPONSE: The text was corrected.

  1. Page 15: "rearded" please check this word

RESPONSE: The text was changed as follows: Levan nanoparticles also can be used as drug carriers for peptide and proteins [295].

  1. Page 15:  "Sulfated levan promising for future use in cardiac tissue engineering." please check English

RESPONSE: The text was changed as follows: «Sulfated levan can be considered as a promising material for cardiac tissue engineering applications».

  1. Page 15: "with PCL and PEO" Please define

RESPONSE: Abbreviations defined: «with polycaprolactone (PCL) and polyethyleneoxide (PEO)»

  1. Page 16: "Liyanapathiranage et al. analyzed 1,740 complete genome sequences belonging to 39 Xanthomonas spp. retrieved from the National Center for Biotechnology Information (NCBI), a large proportion of them being from the species: X. oryzae (396 genomes), X. citri (195 genomes), X. phaseoli (97 genomes), X. perforans (151 genomes), and X. arboricola (136 genomes) [320]." What were the main findings of this work?

RESPONSE: The main findings of this work added to the manuscript: «The authors studied the genetic organization and patterns of evolution of several clusters of the type VI secretion system (T6SS) in order to understand the contribution of T6SS toward ecology and evolution of Xanthomonas spp.»

Round 2

Reviewer 3 Report

No more comments